# Prevalence and predictors of hypoglycemia in older outpatients with type 2 diabetes mellitus

Ahmad Al-Azayzih[1,2]*, Roaa J. Kanaan[2], Shoroq M. Altawalbeh[2], Karem H. Alzoubi[1,2], Zelal Kharaba[1,3], Anan Jarab[2,4]

1 Department of Pharmacy Practice and Pharmacotherapeutics, College of Pharmacy, University of Sharjah, Sharjah, United Arab Emirates, 2 Department of Clinical Pharmacy, Faculty of Pharmacy, Jordan University of Science and Technology, Irbid, Jordan, 3 Faculty of Medical Sciences, Newcastle University, Newcastle upon Tyne, United Kingdom, 4 Department of Clinical Pharmacy, College of Pharmacy, Al Ain University, Abu Dhabi Campus, Abu Dhabi, United Arab Emirates

* aaazayzih@just.edu.jo

**Data Availability Statement:** All relevant data are within the manuscript and its Supporting information files.

## Abstract

### Background

The prevalence of type 2 diabetes (DM) has been increasing globally, particularly among older adults who are more susceptible to DM-related complications. Elderly individuals with diabetes are at higher risk of developing hypoglycemia compared with younger diabetes patients. Hypoglycemia in elderly patients can result in serious consequences such as cognitive changes, increased risk of falls, heart and other vascular problems, and even high mortality rate.

### Objective

To assess prevalence, and factors associated with hypoglycemia events among geriatric outpatients with type 2 diabetes mellitus.

### Methods

The study was conducted at King Abdullah University Hospital (KAUH) at the outpatient diabetes clinic from October 1st, 2022 to August 1st, 2023. Variables such as socio-demographics, medication history, and comorbidities were obtained using electronic medical records. The prevalence of hypoglycemia was determined through patient interviews during their clinic visit. Patients were prospectively monitored for hospital admissions, emergency department visits, and mortality using electronic medical records over a three-month follow-up period. Logistic regression models were conducted to identify factors associated with hypoglycemia and hospital admissions/ emergency visits. Ethical Approval (Reference # 53/151/2022) was obtained on 19/9/2022.

**Funding:** The author(s) received no specific funding for this work.

**Competing interests:** The authors have declared that no competing interests exist.

## Results

Electronic medical charts of 640 patients who have type 2 diabetes mellitus and age $\geq 60$ years were evaluated. The mean age ± SD was 67.19 (± 5.69) years. Hypoglycemia incidents with different severity levels were prevalent in 21.7% (n = 139) of the patients. Insulin administration was significantly associated with more hypoglycemic events compared to other antidiabetic medication. Patients with liver diseases had a significantly higher risk of hypoglycemia, with odds 7.43 times higher than patients without liver diseases. Patients with dyslipidemia also had a higher risk of hypoglycemia (odd ratio = 1.87). Regression analysis revealed that hypoglycemia and educational level were significant predictors for hospital admission and emergency department (ER) visits. Hypoglycemia was a positive predictor, meaning it increased the odds of these outcomes, while having a college degree or higher was associated with reduced odds of hospital admission and ER visits.

## Conclusion

Current study identified a considerable prevalence of hypoglycemia among older patients with type 2 diabetes, particularly, among those with concurrent liver diseases and dyslipidemia. Furthermore, hypoglycemia was associated with an increased rate of emergency department visits and hospital admissions by 2 folds in this population.

## Introduction

The primary goal of type 2 diabetes management is to maintain normal level of blood glucose (normoglycemia), thus reducing type 2 diabetes-associated health complications including both microvascular and macrovascular problems as well as to improve patients' health related quality of life (HRQoL) [1, 2]. However, maintaining tight control of blood glucose level is commonly associated with elevated risk of hypoglycemic events [3]. Hypoglycemia is defined as an abnormal low blood glucose level in individuals, usually when blood glucose levels drop below 70 mg/dL (3.9 mmol/L) [1]. Hypoglycemia events are generally developed from intensive use of insulin and certain other antidiabetic medications [3], and presence of certain medical conditions such as liver and renal diseases [4, 5].

Hypoglycemia is for certain associated with significant negative consequences among patients with diabetes impacting their cognitive capabilities [6, 7]. Moreover, hypoglycemia poses an obstacle for optimum control of diabetes, as the patient's concerns of developing more hypoglycemic episodes would lead to avoidance of taking their insulin and other antihyperglycemic drugs [8]. Also, left untreated hypoglycemia was found to be associated with increased incidence of cardiovascular diseases, hospital admissions, and mortality [9, 10].

Older diabetic patients are more prone to experiencing recurrent hypoglycemic episodes compared to younger patients due to several factors [11]. Firstly, elderly diabetic patients are more likely to present with decline in their physiological functions, resulting in variable response of counterregulatory hormones during hypoglycemic events [12]. Secondly, the complexity of medications regimens and presence of co-existed chronic conditions make it challenging to avoid hypoglycemic events in those patients [11, 13]. Finally, lifestyles factors such as irregular eating habits, decreased appetite, and malnutrition are commonly prevalent among elderly patients which are significantly associated with hypoglycemic occurrence and severity [14].

Despite the well-documented correlation between type I diabetes management and hypo-glycemia occurrences, there is still limited reports regarding the actual prevalence, incidence, and predictors of hypoglycemic events among patient with type 2 diabetes, especially the older patients. Older patients with type 2 diabetes who encounter hypoglycemia adverse events tend to seek more frequent visits to emergency department and health care facilities [15].

There has been a scarcity of research investigating prevalence, incidence, and predictors of hypoglycemia among elderly patients with type 2 diabetes in Jordan. Therefore, this study aims were to assess the prevalence of hypoglycemic events and assess the potential factors contributing to their occurrence in older individuals diagnosed with type 2 diabetes mellitus.

## Methodology

### Study design and settings

To examine the primary objective of identifying the prevalence and factors associated with hypoglycemia occurrence, a cross-sectional study was conducted at the outpatient diabetes clinic of King Abdullah University Hospital (KAUH) over a ten-month period spanning from October 1st, 2022, to August 1st, 2023. Inclusion criteria included individuals aged 60 years and above with a confirmed diagnosis of type 2 diabetes mellitus (DM) who were treated with at least one antidiabetic medication. The prevalence of and severity of hypoglycemia were determined through patient interviews utilizing a structured questionnaire designed for this purpose. Utilizing electronic medical records, patients were prospectively monitored for hospital admissions, visits to the emergency department, and mortality over a three-month follow-up period, adhering to the framework of a prospective cohort study design.

### Patients characteristics and data collection

Patients characteristics including age, gender, marital status, occupation, monthly income, smoking status, educational level, physical activity family history of DM, and current chronic diseases status in study subjects were obtained from the patients during the interview and from hospital medical charts. Glycated hemoglobin (HbA1c) blood levels, as well as medication information including past and present medications, were obtained from the hospital's electronic medical records. Data about hypoglycemia during the three months preceding the clinic visit were obtained from the patients during the interview and validated using medical records and lab results. Hypoglycemia severity levels were assessed based on both American Diabetes Association and The International Hypoglycemia Study Group recommendations [16]. Hospital admissions, emergency department visits frequency and reasons were obtained from the electronic records during the three-months period post the interview. The interviews lasted around 10–15 minutes. Charleson comorbidity index was calculated for each patient to indicate morbidity status and to estimate the risk of death from comorbid diseases. Missing data were excluded from the analysis to ensure the integrity and reliability of the results.

### Sample size calculation

For sample size calculation, the following formula *(Sample size (n) = Z² P(1-P)/d²)* was used to find the minimum number of individuals required for achieving a 95% confidence interval (z = 1.96) with absolute error of 5% (d = 0.05), and with assumed proportion in population of 50% (P = 0.5) to ensure maximum number of individuals included in this study [17]. To meet the necessary sample size for this study, 385 patients or more needed to be included.

## Data analysis

Data was analyzed using Stata version 17 software (Stata Crop. 2021). Continuous variables such as age, HbA1c, and number of comorbidities were summarized in terms of their mean and standard deviation. Categorical variables such as gender, body mass index, and marital status were described using frequencies and percentages. Backward stepwise logistic regression models with P value < 0.2 to stay were conducted to evaluate factors associated with hypoglycemia in the period of three months preceding the clinic visit and hospital admission/ emergency visit during the follow up period. A two-sided P value of less than 0.05 was assumed statistically significant.

## Ethical approval and informed consent

Approval for this study was obtained from the Institutional Review Board of KAUH at Jordan University of Science and Technology (Ref. # 53-151-2022, Approval date: 19/9/2022). All interviewed patients in this study have completed written informed consent forms for the purpose of participation.

## Results

A total of 640 patients with type 2 diabetes aged 60 years and above were recruited in the study, with a mean age ± SD of 67.19 (± 5.69) years. Hypoglycemia was prevalent in 21.7% of the patients. Of the total subjects, 55.78% were females, 82.81% were married, 15.9% were living alone, 84.22% were unemployed or retired. A total of 52.34% of the patients had a low monthly income, earning less than 500 JOD per month. Only 25.16% of the participants were smokers, and 29.06% were physically active. Additionally, 55.94% of the patients had a family history of diabetes mellitus. A statistically significant difference was identified between the groups with or without hypoglycemia regarding the family history of diabetes (p = 0.003). (Table 1).

This study showed that 14.4% (n = 20) of the elderly patients with type 2 diabetes experienced severe level of hypoglycemia (level 3), while 42.4% (n = 59) of the patients experienced moderate level of hypoglycemia (level 2), and 43.2% (n = 60) experienced mild level of hypoglycemic events (level 1). (Fig 1).

The results indicated that the mean number of comorbid conditions was 2.72 (SD = 1.47), with hypertension being the most common (79.22%), followed by dyslipidemia (58.13%) and myocardial infarction (16.09%). On average, patients were taking 6.59 (SD = 2.72) medications, including insulin, oral antihyperglycemic agents, and other drugs to treat comorbid conditions. The average number of medications indicated for diabetes mellitus type 2 was 1.70 (SD = 0.87). Additionally, the mean value for HbA1C among patients who experienced hypoglycemic episodes was 7.82% (SD = 1.45) while it was 7.73 (SD = 1.81) among those patients who did not experience any hypoglycemic episodes during the 3 months follow up period. Additionally, the mean value of the Charlson comorbidity index was 3.83 (± 0.91), indicating a moderate severity of comorbid diseases. Among the study sample, 17.97% had hospital admissions during the three months follow up period, while emergency department visits were experienced by 8.13% of patients. (Tables 2 and 3).

Our results indicated that the most commonly prescribed antidiabetic medication was metformin followed by sulfonylurea medications (Glimepiride, Gliclazide, Glibenclamide). Metformin was administered in various strengths (500, 850, or 1000 mg) to 501 individuals. Notably, only 107 patients who were taking metformin alone or in combination with other oral hypoglycemic agents experienced hypoglycemia and the difference was not statistically significant compared to the non-hypoglycemia group. Furthermore, among the 264 patients

**Table 1. Demographic characteristics of the study patients.**

| Variable | | Total population (n = 640) | | | |
|---|---|---|---|---|---|
| | | Mean (±SD) or n (%)[a] | Number of patients with at least one hypoglycemic episode ** (n = 139) 21.7% | Number of patients with no hypoglycemic episodes ** (n = 501) 78.3% | P-value |
| | | | Mean (±SD) or n (%)[a] | Mean (±SD) or n (%)[a] | |
| Age | | 67.19 (±5.69) | 66.74 (±5.51) | 67.31 (±5.73) | 0.296 |
| Gender | Male | 283 (44.22) | 48 (16.96) | 235 (83.04) | |
| | Female | 357 (55.78) | 91 (25.49) | 266 (74.51) | 0.009 |
| Body mass index (kg/m2) | Normal | 112 (17.50) | 29 (25.89) | 83 (74.11) | 0.621 |
| | Overweight | 233 (36.40) | 49 (21.03) | 184 (78.97) | |
| | Obese | 295 (46.10) | 62 (21.02) | 233 (78.98) | |
| Marital status | Married | 530 (82.81) | 115 (21.70) | 415 (78.30) | 0.182 |
| | Single/other | 110 (17.19) | 32 (29.10) | 78 (70.90) | |
| Living conditions | Not living alone | 538 (84.10) | 122 (22.68) | 416 (77.32) | 0.586 |
| | Living alone | 102 (15.90) | 42 (41.18) | 60 (58.82) | |
| Residency | City | 311 (48.59) | 61 (19.61) | 250 (80.39) | 0.225 |
| | Countryside | 329 (51.41) | 78 (20.53) | 302 (79.47) | |
| Educational level | College/university | 260 (40.63) | 61 (23.46) | 199 (76.54) | 0.376 |
| | Less than college | 380 (59.38) | 56 (17.50) | 264 (82.50) | |
| Job Status | Retired/Unemployed | 539 (84.22) | 127 (23.56) | 412 (76.44) | 0.172 |
| | Employed | 101 (15.78) | 38 (37.62) | 63 (62.38) | |
| Monthly income | More than 1000 JD* | 147 (22.97) | 41 (27.89) | 106 (72.11) | 0.110 |
| | 500–1000 JD* | 158 (24.69) | 33 (20.89) | 125 (79.11) | |
| | Less than 500 JD* | 335 (52.34) | 65 (19.40) | 270 (80.60) | |
| Smoking status | | 161 (25.16) | 38 (23.60) | 123 (76.40) | 0.503 |
| Physical activities | | 186 (29.06) | 44 (23.66) | 142 (76.34) | 0.447 |
| Family history of Diabetes Miletus | | 358 (55.94) | 93 (25.98) | 265 (74.02) | 0.003 |

*Jordanian dinar; SD: Standard deviation;

**hypoglycemia within the previous three months.

receiving sulfonylureas alone or in combination, 54 individuals encountered hypoglycemia, indicating no statistical difference between hypoglycemia and non-hypoglycemia group. However, among the 243 patients prescribed insulin (glargine, detemir, degludec, Aspart, regular, NPH), 65 patients experienced hypoglycemia, and this association was found to be statistically significant (p = 0.016). For DPP4 inhibitors (saxagliptin and linagliptin) administered to 116 patients, 31 experienced hypoglycemia. Of the 3 patients treated with GLP1 agonists (liraglutide and semaglutide), only 1 patient experienced hypoglycemia. Furthermore, SGLT2-I was administered to 40 patients, with 10 experienced hypoglycemia, and no statistically significant difference was identified between the groups. (Table 4).

Multivariate logistic regression analysis was applied to examine the potential predictors associated with hypoglycemia. Results showed that patients with a history of liver diseases exhibit 7.43 times higher odds of experiencing hypoglycemia compared to those without liver diseases (P = 0.020). Furthermore, dyslipidemia as a predictor reveals an odds ratio of 1.87, suggesting an 87% elevated likelihood of hypoglycemia (P = 0.032). Similarly, insulin use has

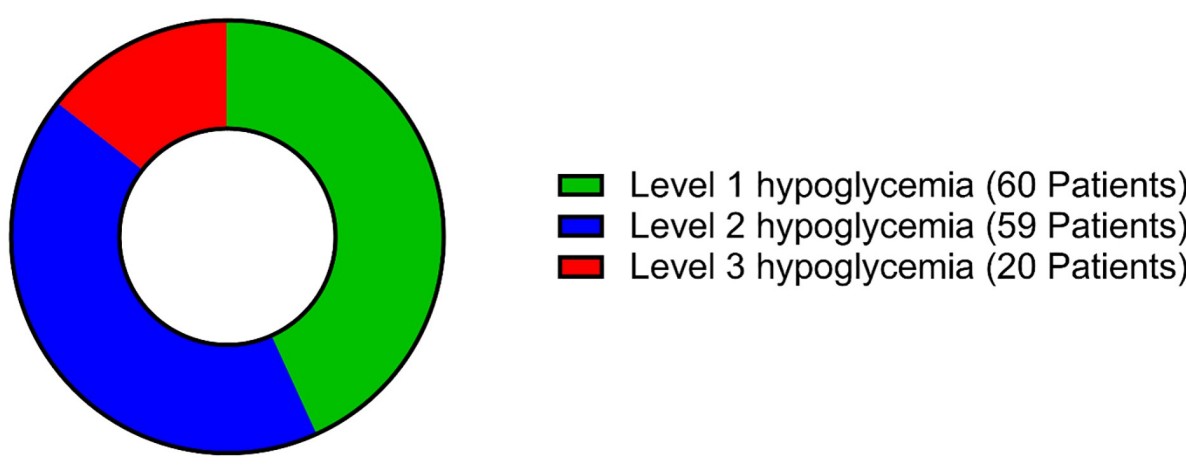

**Total=139**

**Fig 1. Patients distribution according to hypoglycemia severity levels (n = 139).** Level 1 hypoglycemia: Blood glucose level of 70 mg/dL or less. Level 1 hypoglycemia: Blood glucose level of less than 54 mg/dL. Level 3 hypoglycemia: Involves severe cognitive impairment which needs assistance for recovery [1].

revealed odds of 1.88, meaning that there is 88% increase in the odds of hypoglycemia for individuals receiving insulin (P = 0.015). Moreover, individuals with a family history of diabetes mellitus exhibit 1.82 higher odds of experiencing hypoglycemia episodes compared to those without family history of DM. (P = 0.023). (Table 5).

Results of this study showed that individuals who have experienced hypoglycemia had 1.95 times higher odds of hospital admissions and emergency department (ER) visits compared to those without hypoglycemia episodes. (95% C.I = 1.25–3.04, P = 0.003). Also, Patients who had higher educational level exhibit lower odds (OR = 0.61) compared to patients who did not complete their education, this mean that hospital admissions and ER visits in patients with higher education were lower by 39% compared to patients with lower education level (95% CI = 0.40–0.94). (Table 6).

## Discussion

Type 2 diabetes mellitus is highly prevalent within the population of Jordan and neighboring countries. Our study was conducted to investigate the prevalence of hypoglycemic events among older outpatients diagnosed with type 2 diabetes from Jordan. Additionally, the study aimed to identify the main predictors linked to development of hypoglycemic conditions among elderly diabetic population attending outpatient clinics.

Results of the current study estimated that prevalence of hypoglycemia among elderly patients was 21.7%. Hypoglycemia prevalence among type 2 diabetes patients was reported previously in other studies ranged from 12% to 57.44% [18–20]. Hypoglycemia is generally associated with severe negative consequences including increased risk of cardiac arrhythmias, falls, and even mortality [21, 22]. Our study indicated that 14.4% of the study patients reported severe hypoglycemia, while 42.4 and 43.2% of the patients reported moderate and mild hypoglycemia, respectively. A systematic and meta-analysis population-based study included 46 studies in their analysis had prevalence estimate of hypoglycemia of 45% for mild and

**Table 2. Disease and biomedical characteristics of the study patients.**

| Variables | Total population (n = 640) | | | |
|---|---|---|---|---|
| | Mean (±SD) or n (%)[a] | Number of patients with at least one hypoglycemic episode [1] (n = 139) Mean (±SD) or n (%)[a] | Number of patients with no hypoglycemic episodes [1] (n = 501) Mean (±SD) or n (%)[a] | P-value |
| **Number of other chronic diseases** | 2.72 (±1.47) | 2.60 (±1.38) | 2.76 (±1.49) | 0.273 |
| **Hypertension** | 507 (79.22) | 108 (21.30) | 399 (78.70) | 0.617 |
| **Dyslipidemia** | 372 (58.13) | 90 (24.19) | 282 (75.81) | 0.074 |
| **Chronic kidney disease** | 55 (8.59) | 17 (30.91) | 38 (69.09) | 0.084 |
| **Chronic obstructive pulmonary disease** | 16 (2.50) | 3 (18.75) | 13 (81.25) | 0.771 |
| **Leukemia** | 3 (0.47) | 0 (0.00) | 3 (100.00) | 0.360 |
| **Lymphoma** | 3 (0.47) | 1 (33.33) | 2 (66.67) | 0.625 |
| **Stroke** | 49 (7.66) | 11 (22.45) | 38 (77.55) | 0.897 |
| **Myocardium infarction** | 103 (16.09) | 19 (18.45) | 84 (81.55) | 0.379 |
| **Heart failure** | 48 (7.50) | 10 (20.83) | 38 (79.17) | 0.877 |
| **Peptic ulcer disease** | 9 (1.41) | 2 (22.22) | 7 (77.78) | 0.950 |
| **Cancer**\*\* | 33 (5.16) | 9 (27.27) | 24 (72.73) | 0.427 |
| **Liver disease** | 8 (1.25) | 4 (50.00) | 4 (50.00) | 0.051 |
| **Thyroid dysfunction** | 55 (8.59) | 11 (20.00) | 44 (80.00) | 0.746 |
| **Asthma** | 37 (5.79) | 9 (24.32) | 28 (75.68) | 0.696 |
| **HbA1C Level**[b] | 7.75% (1.74) | 7.82% (1.45) | 7.73% (1.81) | 0.632 |
| **Charlson comorbidity index**[c] | 3.83 (±0.91) | 3.77 (±0.85) | 3.84 (±0.93) | 0.408 |

[1]Hypoglycemia within the previous three months;

[a] Standard deviation;

\*P-value for variables significantly associated with hypoglycemia;

\*\*Solid tumor.

[b] Based on total number 539 patients (110 with hypoglycemic episodes and 429 patients without)

[c] Charlson comorbidity index calculated to assess the severity of comorbid diseases.

moderate hypoglycemia, while 6% prevalence for severe hypoglycemia [19]. This study indicated that 14.4% of the study patients reported severe hypoglycemia.

Current study has evaluated multiple variables as suspected predictors for increased hypoglycemic events among patients. The number of diabetes medications was linked significantly to the hypoglycemia events occurrence. This observation was consistent with previous studies, which have shown a similar association between two or more antidiabetic medications use and elevated risk of hypoglycemia in diabetic patients compared to the use of single antidiabetic medication [23, 24].

Glycated hemoglobin (HbA1c) is a reliable indicator of long-term control of blood glucose in DM patients. The study has shown that the mean level of HbA1c was comparable and elevated in all patients (with and without hypoglycemia). While this indicates poor control of blood glucose among elderly patients in this study, it might to some extent explain the low prevalence of hypoglycemic events in DM patients. Improper antidiabetic selection, low dose or number of antidiabetic medications, and poor medication adherence can lead to poor control of blood glucose expressed through elevated HbA1c readings, and thus, less chance of developing hypoglycemic state [25, 26]. Previous studies examining the association between hypoglycemia prevalence and HbA1C levels showed conflicting results, while one study has

**Table 3. Medication characteristics of the patients and outcomes of the three months follow up period.**

| Variables | Total population (n = 640) | | | | |
|---|---|---|---|---|---|
| | | Mean (±SD) or n (%)* | Number of patients with at least one hypoglycemic episode (n = 139) Mean (±SD) or n (%)* | Number of patients with no hypoglycemic episodes (n = 501) Mean (±SD) or n (%)* | P-value |
| **Diabetes medications frequency** | | | 2.19 (±0.67) | 2.22 (±0.64) | 0.660 |
| | Once | 77 (12.03) | 19 (24.68) | 58 (75.32) | 0.870 |
| | Twice | 351 (54.84) | 75 (21.37) | 276 (78.63) | |
| | Three times and more | 212 (33.13) | 45 (21.05) | 167 (78.95) | |
| **Number of diabetes medications**[1] | | 1.70 (±0.87) | 1.78 (±0.94) | 1.68 (±0.85) | 0.215 |
| **Number of total medications** | | 6.59 (±2.72) | 6.16 (±2.44) | 6.71 (±2.78) | 0.034 |
| **Hospital admissions**[b,1] | | 115 (17.97) | 35 (25.2) | 80 (15.9) | 0.012 |
| **Emergency department visits**[c] | | 52 (8.13) | 14 (10.1) | 38 (7.6) | 0.513 |
| **Number of patients who had hospital admissions or emergency department visits**[1] | | 143 (14.5) | 43 (30.9) | 100 (19.9) | 0.006 |

*Standard deviation;

[a] hypoglycemia: considered when blood glucose is < 70mg/dl.

[1] Variables significantly associated with hypoglycemia.

[b] Hospital admissions reasons for three months follow up period (asthma exacerbations, hypoglycemia, acute kidney injury, acute coronary syndrome, low oxygen saturation, toxic thyroiditis, hypoglycemia, diabetic foot infection, heart failure exacerbation, intracranial hemorrhage, stroke, esophagitis, pulmonary edema, pulmonary fibrosis).

[c] Emergency department visits reasons (Allergy, hypoglycemia, anemia, drowsiness, COVID19, fatigue, flu, hypertensive crises, hemorrhoids, shortness of breath, tachycardia, urinary tract infections, chest pain, diarrhea, peripheral edema, epigastric pain, hyperglycemia and severe headache).

indicated that prevalence of hypoglycemia increases for HbA1c readings below 7.0% [27], the other study found that higher HbA1c levels are associated with increased the odds for hypoglycemia-related hospitalization [28].

Cardiovascular diseases, dyslipidemia, hypothyroidism, and chronic kidney disease were the most common chronic conditions encountered in this study. Numerous chronic conditions including chronic kidney diseases, hypothyroidism, and liver diseases were considered as

**Table 4. Association between specific antidiabetic medication and hypoglycemia occurrence.**

| Medication Category | N (%) | Number of patients with at least one hypoglycemic episode N (%) | Number of patients with no hypoglycemic episodes N (%) | P-value |
|---|---|---|---|---|
| **Biguanides (Metformin)** | 501 (78.28) | 107 (76.98) | 394 (78.64) | 0.674 |
| **Sulfonylurea** | 264 (41.25) | 54 (38.85) | 210 (41.92) | 0.516 |
| **Insulin** | 243 (37.97) | 65 (46.76) | 178 (35.53) | 0.016 |
| **DPP4-I** | 116 (18.13) | 31 (22.30) | 85 (16.97) | 0.148 |
| **GLP-1A** | 3 (0.47) | 1 (0.72) | 2 (0.40) | 0.625 |
| **SGLT2-I** | 40 (6.25) | 10 (7.19) | 30 (5.99) | 0.603 |

DPP4-I: Dipeptidyl peptidase-4 inhibitor, GLP-1A Glucagon-Like Peptide-1 Receptor Agonists, SGLT2-I: Sodium-Glucose Transport Protein 2 (SGLT2) Inhibitors

**Table 5. Predictors of hypoglycemia in elderly patients with type 2 DM.**

| Variable | OR[a] | 95% CI [b] | P-value |
|---|---|---|---|
| **Number of Diabetes medications** | 1.28 | 0.96–1.70 | 0.094 |
| **Gender (Female)** | 1.51 | 0.83–2.73 | 0.176 |
| **Marital status (Married)** | 1.63 | 0.87–3.06 | 0.126 |
| Liver disease ** | 7.43 | 1.38–40.15 | 0.020 |
| **Physical activities** | 1.67 | 0.95–2.91 | 0.073 |
| **Dyslipidemia**** | 1.87 | 1.06–3.30 | 0.032 |
| **Job status (Employed)** | 0.37 | 0.08–1.66 | 0.193 |
| Insulin ** | 1.88 | 1.13–3.13 | 0.015 |
| **Family history of Diabetes Mellitus (with)**** | 1.82 | 1.08–3.05 | 0.023 |
| **Smoking (Smoker)** | 1.77 | 0.90–3.46 | 0.097 |

**variables significantly associated with hypoglycemia;

[a] odds ratio;

[b] confidence interval.

independent risk factors associated with drop of blood glucose and have been linked to elevated hypoglycemic events in diabetic and non-diabetic people [5, 29, 30].

The presence of chronic comorbidities including liver conditions and dyslipidemia were linked to an increase in the risk of hypoglycemic events in this study. Contrarily, chronic kidney disease and hypothyroidism assessment failed to establish a significant association with more hypoglycemic instants. Reports explained the association between liver conditions and raised odds of hypoglycemia due to poor glycogen level and malnutritional status among these patients resulted in existence of hypoglycemia [30]. Dyslipidemia and hypertriglyceridemia were clearly linked to insulin resistance and thus, hyperglycemia [31]. Various pharmacological studies have examined the effects of drugs prescribed for dyslipidemia on the blood glucose level, with findings indicating that the administration of statins and ezetimibe may induce notable hypoglycemia [32, 33]. Consequently, the hypoglycemic events in patients with co-existed dyslipidemia may be attributable to the pharmacological effects of the administered drugs. Furthermore, insulin resistance is associated with dyslipidemia, creating a reciprocal relationship in which each condition has the potential to exacerbate the other [34], thereby

**Table 6. Variables associated with hospital admissions or emergency department visits*.**

| Variable | OR[a] | (95% CI)[b] | P-value |
|---|---|---|---|
| **Hypoglycemia**** | 1.95 | (1.25–3.04) | 0.003 |
| **Gender (Female)** | 0.69 | (0.45–1.06) | 0.090 |
| **Marital status (Married)** | 1.56 | (0.91–2.65) | 0.104 |
| **Education (college/university)**** | 0.61 | (0.40–0.94) | 0.026 |
| **Family history of DM[C] (with)** | 1.23 | (0.82–1.85) | 0.309 |
| **Number of Diabetes medications** | 0.87 | (0.69–1.10) | 0.259 |
| **CCI** | 1.09 | (0.88–1.35) | 0.409 |

*The independent variables included in the regression model;

** variables significantly associated with hospital admissions and emergency department visits;

[a] odd ratio;

[b] confidence interval;

[c] diabetes mellitus.

necessitating mainly the administration of higher insulin doses [35]. This increase in dosage may, in turn, lead to a higher frequency of hypoglycemic episodes.

From another perspective, this study has examined the variables that might be associated with frequent rates of emergency department visits or hospital admissions. In current study, Patients who had developed hypoglycemia were more subjected to emergency department visits and hospital admissions. A previous study has highlighted that recurrent emergency department visits after hypoglycemic events were notably more common in those taking oral antidiabetic agents compared to insulin only [36]. Another study indicated hypoglycemia as a major risk factor for hospital admission among older adults with type 2 diabetes [15].

## Study limitations

This study might be subjected to recall bias due to reliance on patients self-reporting concerning the frequency and severity of hypoglycemic events encountered, even though the study methodology ensured to assess the hypoglycemic events carefully based on structured patients interviews with standardized questions.

## Conclusion

This study revealed a notable prevalence of hypoglycemia events occurrence among elderly outpatients with type diabetes mellitus, particularly patients with concurrent liver diseases and dyslipidemia. Additionally, Hypoglycemia was correlated with a twofold increase in the rate of emergency department visits and hospital admissions among elderly patients with type 2 diabetes. Prescribers should conduct thorough medication assessments to minimize occurrence of hypoglycemic events among older outpatients with type 2 diabetes.

## Supporting information

**S1 File. Coded data for analysis.**
(XLSX)

## Acknowledgments

The research was approved by the Jordan University of Science and Technology (JUST) Research Counsel and Deanship of Research (Approval /Research number is 559–2022).

## Author Contributions

**Conceptualization:** Ahmad Al-Azayzih, Roaa J. Kanaan, Shoroq M. Altawalbeh, Anan Jarab.

**Data curation:** Ahmad Al-Azayzih, Roaa J. Kanaan, Shoroq M. Altawalbeh, Anan Jarab.

**Formal analysis:** Ahmad Al-Azayzih, Roaa J. Kanaan, Shoroq M. Altawalbeh, Karem H. Alzoubi, Zelal Kharaba.

**Investigation:** Ahmad Al-Azayzih, Karem H. Alzoubi, Anan Jarab.

**Methodology:** Ahmad Al-Azayzih, Roaa J. Kanaan, Shoroq M. Altawalbeh, Karem H. Alzoubi, Zelal Kharaba, Anan Jarab.

**Project administration:** Ahmad Al-Azayzih, Shoroq M. Altawalbeh.

**Supervision:** Ahmad Al-Azayzih, Shoroq M. Altawalbeh.

**Validation:** Ahmad Al-Azayzih, Shoroq M. Altawalbeh, Zelal Kharaba.

**Visualization:** Ahmad Al-Azayzih, Anan Jarab.

**Writing – original draft:** Ahmad Al-Azayzih, Roaa J. Kanaan, Shoroq M. Altawalbeh, Karem H. Alzoubi, Zelal Kharaba, Anan Jarab.

**Writing – review & editing:** Ahmad Al-Azayzih, Roaa J. Kanaan, Shoroq M. Altawalbeh, Karem H. Alzoubi, Zelal Kharaba, Anan Jarab.

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
