## [Decision Letter · Decision Letter 0]

8 Apr 2024

PONE-D-24-05623Prevalence and predictors of hypoglycemia in older adult patients with type 2 diabetes mellitusPLOS ONE

Dear Dr. Al-Azayzih,

Thank you for submitting your manuscript to PLOS ONE. After careful consideration, we feel that it has merit but does not fully meet PLOS ONE’s publication criteria as it currently stands. Therefore, we invite you to submit a revised version of the manuscript that addresses the points raised during the review process.

We look forward to receiving your revised manuscript.

Kind regards,

Maher Abdelraheim Titi

Academic Editor

PLOS ONE

Journal Requirements:

2. Thank you for submitting the above manuscript to PLOS ONE. During our internal evaluation of the manuscript, we found significant text overlap between your submission and previous work in the [introduction, conclusion, etc.].

Please revise the manuscript to rephrase the duplicated text, cite your sources, and provide details as to how the current manuscript advances on previous work. Please note that further consideration is dependent on the submission of a manuscript that addresses these concerns about the overlap in text with published work.

[If the overlap is with the authors’ own works: Moreover, upon submission, authors must confirm that the manuscript, or any related manuscript, is not currently under consideration or accepted elsewhere. If related work has been submitted to PLOS ONE or elsewhere, authors must include a copy with the submitted article. Reviewers will be asked to comment on the overlap between related submissions (http://journals.plos.org/plosone/s/submission-guidelines#loc-related-manuscripts).]

We will carefully review your manuscript upon resubmission and further consideration of the manuscript is dependent on the text overlap being addressed in full. Please ensure that your revision is thorough as failure to address the concerns to our satisfaction may result in your submission not being considered further.

3. We note that your Data Availability Statement is currently as follows: [All relevant data are within the manuscript and its Supporting Information files]

Reviewers' comments:

Reviewer's Responses to Questions

**Comments to the Author**

1. Is the manuscript technically sound, and do the data support the conclusions?

Reviewer #1: Partly

Reviewer #2: No

2. Has the statistical analysis been performed appropriately and rigorously? 

Reviewer #1: No

Reviewer #2: N/A

3. Have the authors made all data underlying the findings in their manuscript fully available?

Reviewer #1: Yes

Reviewer #2: Yes

4. Is the manuscript presented in an intelligible fashion and written in standard English?

Reviewer #1: Yes

Reviewer #2: Yes

5. Review Comments to the Author

Reviewer #1: My comments are in the attached file. The sample size calculation should be clearly stated. The standard deviation for the mean total cholesterol and LDL-c is quite large. The author need to look at the data again. Discussion need to be re-done

Reviewer #2: Prevalence and predictors of hypoglycemia in older adult patients with type 2 diabetes mellitus

Review:

The authors have tried to study hypoglycemia episodes among elderly patients. There are several queries that arise in our mind when we go through the study methodology:

1. The authors have mentioned that it is a cross sectional study. But they also say, they followed up OP patients for 3 months. Will this not come under a prospective cohort study? Thus, the hypoglycemia that they mention will be incidence and not prevalence.

2. There is no clear mention of when the electronic data of the patients were assessed – was it the baseline data at the first visit or was it during the admission later. Were the patient’s who were interviewed initially and that agreed to study were followed up for next 3 months or other patients who did not agree to participate in the study but were admitted for hypoglycemia were included?? Not clear from the manuscript

3. In the limitations – the authors have mentioned about recall bias; it is unclear why recall bias when the source is electronic data

4. The sample size calculation – the basis needs to be explained.

5. When we say hypoglycemia prevalence among geriatric population with type 2 diabetes mellitus, it should include all elders with diabetes. When we include a hospital-based study among patients attending OP or IP for one or other health issues, it might not be representative of all elders with diabetes.

6. Further, in this study for prevalence of hypoglycemia, they are considering only patients who are getting admitted, visit emergency or die in the next 3 months period. Will it not include only the severe cases and leave the milder cases of hypoglycemia? Can we call as prevalence of severe hypoglycemia among elderly with diabetes?

7. In table 4, they mention about hospital admissions as 115 and emergency visits as 52 and both as 112. Is it an arithmetic error, again not clear?

8. In the tables 2- 4, it is given as hypoglycemia episodes, is it no of episodes or no of patients with hypoglycemia?

9. In table 4, what is the denominator for the percentages calculated? It should be out of the no. of episodes? Here, it is neither no. of episodes nor no. of hospital admissions.

6. PLOS authors have the option to publish the peer review history of their article (what does this mean?). If published, this will include your full peer review and any attached files.

Reviewer #1: No

Reviewer #2: No

---

## [Author Response · Author response to Decision Letter 0]

23 May 2024

Response letter to the Reviewers’ Comments. 

Reviewer #1: My comments are in the attached file. The sample size calculation should be clearly stated. The standard deviation for the mean total cholesterol and LDL-c is quite large. The author need to look at the data again. Discussion need to be re-done.

Would like to thank the reviewer for his/her valid comments. All comments were addressed in the main manuscript. Actually, the authors had rewritten the most part of the manuscript including introduction, methodology..etc. 

Sample size calculation formula and details were addressed in the methodology part. 

The authors have removed table number 4 which includes the biomedical data because of the issue of missing data and since we believe that it will not affect the essence and main conclusion of the manuscript. 

“Cardiovascular diseases, dyslipidemia, hypothyroidism, and chronic kidney disease were the most common chronic conditions encountered in this study These were not stated in the results. There is no thyroid function test result or frequency of hypothyroidism. There is nothing about eGFR so CKD cannot be discussed.”

These comorbidities do co-exist among the study participants and they have been diagnosed with these conditions already. Please view table number 2 to view the frequency of these conditions. 

Reviewer#2: 

1. The authors have mentioned that it is a cross sectional study. But they also say, they followed up OP patients for 3 months. Will this not come under a prospective cohort study? Thus, the hypoglycemia that they mention will be incidence and not prevalence.

Would like to thank the reviewer for his/her concern. 

For the primary objective of measuring the prevalence and factors associated with hypoglycemia, a Cross-sectional study was conducted at King Abdullah University Hospital (KAUH) at the outpatient diabetes clinic over 10 months period, from October 1st, 2022 to August 1st, 2023. Over a three-month follow-up period, electronic medical records were utilized to prospectively track recruited patients for hospital admissions, emergency department visits, and mortality within the framework of a prospective cohort study design.

This issue has been clarified in the text.

2. There is no clear mention of when the electronic data of the patients were assessed – was it the baseline data at the first visit or was it during the admission later. Were the patient’s who were interviewed initially and that agreed to study were followed up for next 3 months or other patients who did not agree to participate in the study but were admitted for hypoglycemia were included?? Not clear from the manuscript.

Electronic data were assessed, and patients were interviewed at their first visit to the clinic. The same patients interviewed were followed up for the next 3 months for any emergency/hospital visits. 

This issue has been clarified in the text.

3. In the limitations – the authors have mentioned about recall bias; it is unclear why recall bias when the source is electronic data.

First of All, apologize for any misunderstanding. We are here talking about hypoglycemia occurrences (if it happened for example, how many times, and blood glucose reported at that time). This question was answered through interviewing the patients and this might lead to a recall bias which could be associated with self-reporting. 

This issue has been clarified in the text.

4. The sample size calculation – the basis needs to be explained.

Please view the methodology section for clarification. 

5. When we say hypoglycemia prevalence among geriatric population with type 2 diabetes mellitus, it should include all elders with diabetes. When we include a hospital-based study among patients attending OP or IP for one or other health issues, it might not be representative of all elders with diabetes.

Agree with the reviewer on this. We have changed the title and any relevant data to focus on outpatient older patients type 2 DM patients. 

Our future work would focus on the inpatient older subjects suffering from type 2 DM. 

6. Further, in this study for prevalence of hypoglycemia, they are considering only patients who are getting admitted, visit emergency, or die in the next 3 months period. Will it not include only the severe cases and leave the milder cases of hypoglycemia? Can we call as prevalence of severe hypoglycemia among elderly withdiabetes?

Hypoglycemia was assessed during the interview (self-report) in the initial clinic visits. Readmissions and ED visits assessed were all-cause visits occurred during the three months follow up period. So we did not follow patients particularly for hypoglycemia. The main objective of this study was to assess the prevalence of hypoglycemia. As a secondary objective, we have evaluated the association between hypoglycemia (self-reported during the clinic visits), along with other potential predictors, and the occurrence of hospital admissions/emergency department visits during the follow up period.

7. In table 4, they mention hospital admissions as 115 and emergency visits as 52 and both as 112. Is it an arithmetic error, again not clear?

Patients how had hospital admissions or emergency department visits were 143. These numbers are actually number of patients not visits. Some patients had both, so it is expected to have the number of patients how had hospital admissions or emergency department visits to be lower than (115+52). The data are presented in table 3

8. In the tables 2- 4, it is given as hypoglycemia episodes, is it no of episodes or no of patients with hypoglycemia?

These are the number of patients who had at least one episode of hypoglycemia during the three months preceding the clinic visit.

9. In table 4, what is the denominator for the percentages calculated? It should be out of the no. of episodes. Here, it is neither no. of episodes nor no. of hospital admissions.

The denominator is the number of patients. For example: 501 patients were on Metformin (out of 640; 78.3%). 394 of them were among the group who did not report any hypoglycemic episode (out of 501 patients; 78.6%). 107 of them were among the group who did report at least one hypoglycemic episode (out of 139 patients; 76.98% [ corrected in the manuscript).

We apologize for mistakenly reporting row percentages rather than the column percentages. All percentages are now corrected, and column percentages are reported.

---

## [Decision Letter · Decision Letter 1]

24 Jul 2024

PONE-D-24-05623R1Prevalence and predictors of hypoglycemia in older outpatients with type 2 diabetes mellitusPLOS ONE

Dear Dr. Al-Azayzih,

Thank you for submitting your manuscript to PLOS ONE. After careful consideration, we feel that it has merit but does not fully meet PLOS ONE’s publication criteria as it currently stands. Therefore, we invite you to submit a revised version of the manuscript that addresses the points raised during the review process.

We look forward to receiving your revised manuscript.

Kind regards,

Maher Abdelraheim Titi

Academic Editor

PLOS ONE

Journal Requirements:

Reviewers' comments:

Reviewer's Responses to Questions

**Comments to the Author**

1. If the authors have adequately addressed your comments raised in a previous round of review and you feel that this manuscript is now acceptable for publication, you may indicate that here to bypass the “Comments to the Author” section, enter your conflict of interest statement in the “Confidential to Editor” section, and submit your "Accept" recommendation.

Reviewer #3: All comments have been addressed

Reviewer #4: All comments have been addressed

2. Is the manuscript technically sound, and do the data support the conclusions?

Reviewer #3: Yes

Reviewer #4: Partly

3. Has the statistical analysis been performed appropriately and rigorously? 

Reviewer #3: Yes

Reviewer #4: Yes

4. Have the authors made all data underlying the findings in their manuscript fully available?

Reviewer #3: Yes

Reviewer #4: Yes

5. Is the manuscript presented in an intelligible fashion and written in standard English?

Reviewer #3: Yes

Reviewer #4: Yes

6. Review Comments to the Author

Reviewer #3: This article deals with an important issue - hypoglycaemia in diabetes mellitus. The design, materials and methods, as well as the results, are classically structured. The examination reveals a number of points:

1. add the date and number of the ethics committee in the abstract

2. add type of diabetes mellitus to the keywords

3. in the text also specify the number and date of the ethics committee minutes

4. in the list of references, sources older than 5 years, where it is possible to review

5. in the table the designation of glycated haemoglobin HBA1C, preferably use the correct abbreviation HbA1c

Reviewer #4: The paper ** Prevalence and predictors of hypoglycemia in older outpatients with type 2 diabetes mellitus**was reviewed. The results revealed the study identified a considerable prevalence of hypoglycemia among older patients with

type 2 diabetes, particularly, among those with concurrent liver diseases and dyslipidemia. However, there are some concerns necessary to be clarified.

The major concern is that why dyslipidemia is associated with hypoglycemia? The discussion should mention and provide some explanations about the results- treatments associated? Or some confounding factors such as the patients with dyslipidemia treated by insulin.

7. PLOS authors have the option to publish the peer review history of their article (what does this mean?). If published, this will include your full peer review and any attached files.

Reviewer #3: **Yes: **Mariia V. Matveeva

Reviewer #4: **Yes: **Chien-Liang Chen

---

## [Author Response · Author response to Decision Letter 1]

27 Jul 2024

Response to Reviewers’ Comments to the Author

Reviewer #3: This article deals with an important issue - hypoglycemia in diabetes mellitus. The design, materials and methods, as well as the results, are classically structured. The examination reveals several points:

1. Add the date and number of the ethics committee in the abstract:

The date and number of the ethical committee decision was added as recommended.

2. Add type of diabetes mellitus to the keywords

Type 2 diabetes mellitus was added to the keywords part 

3. In the text also specify the number and date of the ethics committee minutes

The date and number of the ethical committee decision was added as recommended under the methodology section.

4. In the list of references, sources older than 5 years, where it is possible to review

First, we would like to thank the reviewer for her valuable feedback on our manuscript. We appreciate your attention to the references included in our study.

1. References Highlighting Original Reports: We have ensured that the references include those that first reported significant findings related to our topic. These original studies are crucial as they lay the groundwork for subsequent research and provide the foundational context for our study. We have marked these original studies in the reference list to highlight their importance. (REF#: 12, 27, 29, and 33)

2. Comprehensive Inclusion of Prevalence Studies: In addition to the original reports, we have included a range of studies that discuss the prevalence of hypoglycemia. Our intention was to provide a comprehensive overview of the existing literature, capturing the various findings and perspectives across different studies. We believe this approach offers a more holistic view of the research landscape. (REF#: 18)

3. Updating References: We have recently updated some references to ensure the most current and relevant studies are cited, reflecting the latest advancements and data in the field. In doing so, we removed older references that have been superseded by more recent work. This was done to maintain the accuracy and relevance of our literature review. Please view the reference list for more details 

5. In the table the designation of glycated haemoglobin HBA1C, preferably use the correct abbreviation HbA1c

The glycosylated Hemoglobin HbA1c was replaced by the Glycated Hemoglobin (HbA1c).

Reviewer #4: The paper ** Prevalence and predictors of hypoglycemia in older outpatients with type 2 diabetes mellitus**was reviewed. The results revealed the study identified a considerable prevalence of hypoglycemia among older patients with

type 2 diabetes, particularly, among those with concurrent liver diseases and dyslipidemia. However, there are some concerns necessary to be clarified.

The major concern is that why dyslipidemia is associated with hypoglycemia? The discussion should mention and provide some explanations about the results- treatments associated? Or some confounding factors such as the patients with dyslipidemia treated by insulin.

The following paragraph was added to the discussion part:

“Various pharmacological studies have examined the effects of drugs prescribed for dyslipidemia on the blood glucose level, with findings indicating that the administration of statins and ezetimibe may induce notable hypoglycemia [32, 33]. Consequently, the hypoglycemic events in patients with co-existed dyslipidemia may be attributable to the pharmacological effects of the administered drugs. Furthermore, insulin resistance is associated with dyslipidemia, creating a reciprocal relationship in which each condition has the potential to exacerbate the other [34], thereby necessitating mainly the administration of higher insulin doses [35]. This increase in dosage may, in turn, lead to a higher frequency of hypoglycemic episodes.”

---

## [Decision Letter · Decision Letter 2]

13 Aug 2024

Prevalence and predictors of hypoglycemia in older outpatients with type 2 diabetes mellitus

PONE-D-24-05623R2

Dear Dr. Al-Azayzih,

We’re pleased to inform you that your manuscript has been judged scientifically suitable for publication and will be formally accepted for publication once it meets all outstanding technical requirements.

Kind regards,

Miquel Vall-llosera Camps

Senior Staff Editor

PLOS ONE

Reviewers' comments:

Reviewer's Responses to Questions

**Comments to the Author**

1. If the authors have adequately addressed your comments raised in a previous round of review and you feel that this manuscript is now acceptable for publication, you may indicate that here to bypass the “Comments to the Author” section, enter your conflict of interest statement in the “Confidential to Editor” section, and submit your "Accept" recommendation.

Reviewer #3: All comments have been addressed

Reviewer #4: All comments have been addressed

2. Is the manuscript technically sound, and do the data support the conclusions?

Reviewer #3: Yes

Reviewer #4: Yes

3. Has the statistical analysis been performed appropriately and rigorously? 

Reviewer #3: Yes

Reviewer #4: Yes

4. Have the authors made all data underlying the findings in their manuscript fully available?

Reviewer #3: Yes

Reviewer #4: Yes

5. Is the manuscript presented in an intelligible fashion and written in standard English?

Reviewer #3: Yes

Reviewer #4: Yes

6. Review Comments to the Author

Reviewer #3: All the questions that were pointed out in the previous review have been covered and answered in full by the authors. There are no new comments. Thank you!

Reviewer #4: All comments and my questions have been reasonally answered.

The paper could provide some clincal evidence to the field of clinical medicine.

7. PLOS authors have the option to publish the peer review history of their article (what does this mean?). If published, this will include your full peer review and any attached files.

Reviewer #3: **Yes: **Mariia Matveeva

Reviewer #4: **Yes: **Chien-Liang Chen, MD

---

## [Editor Report · Acceptance letter]

19 Aug 2024

PONE-D-24-05623R2 

PLOS ONE

Dear Dr. Al-Azayzih, 

I'm pleased to inform you that your manuscript has been deemed suitable for publication in PLOS ONE. Congratulations! Your manuscript is now being handed over to our production team.

Kind regards, 

on behalf of

Dr. Miquel Vall-llosera Camps 

Staff Editor

PLOS ONE